# GlyphControl: Glyph Conditional Control for Visual Text Generation

Yukang Yang[1♮†]     Dongnan Gui[2†♮]     Yuhui Yuan[3†‡]
Weicong Liang[3♮]   Haisong Ding[3]   Han Hu[3]   Kai Chen[3]

[1]Princeton University
[2]University of Science and Technology of China
[3]Microsoft Research Asia

 https://github.com/AIGText/GlyphControl-release

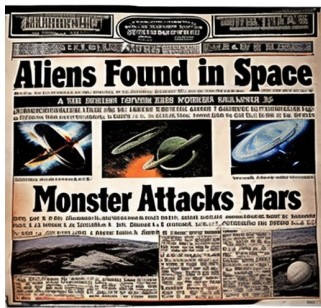

*Newspaper with the headline "Aliens Found in Space" and "Monster Attacks Mars".*

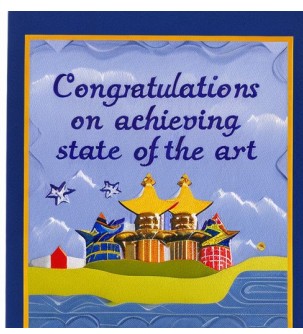

*A decorative greeting card that reads "Congratulations on achieving state of the art".*

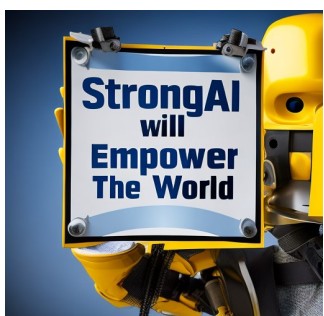

*Dslr portrait of a robot holds a sign that says "StrongAI will Empower The World".*

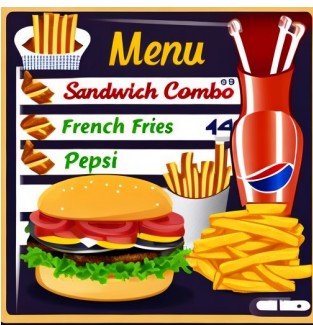

*A menu of a fast food restaurant that contains "Sandwich Combo", "French Fries", and "Pepsi".*

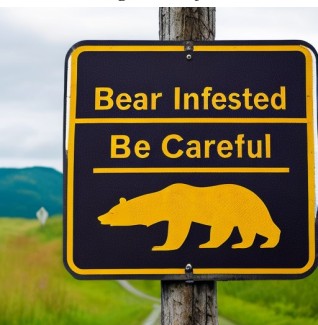

*A sign in front of a beautiful village that says "Bear Infested Be Careful".*

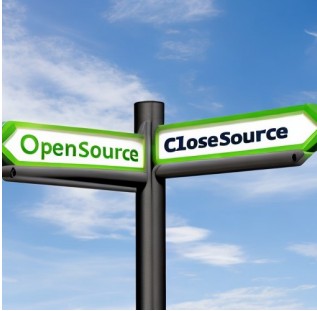

*A sign "OpenSource" facing another sign "CloseSource". They point to two completely different paths.*

Figure 1: Illustrating selected $512 \times 512$ GlyphControl samples for different text prompts and glyph conditions. Our GlyphControl can generate coherent images with well-formed visual text.

## Abstract

Recently, there has been an increasing interest in developing diffusion-based text-to-image generative models capable of generating coherent and well-formed visual text. In this paper, we propose a novel and efficient approach called GlyphControl to address this task. Unlike existing methods that rely on character-aware text

---

[†]Core contribution. [♮] Interns at Microsoft Research Asia
[‡]Corresponding author: yuhui.yuan@microsoft.com

37th Conference on Neural Information Processing Systems (NeurIPS 2023).

encoders like ByT5 and require retraining of text-to-image models, our approach leverages additional glyph conditional information to enhance the performance of the off-the-shelf Stable-Diffusion model in generating accurate visual text. By incorporating glyph instructions, users can customize the content, location, and size of the generated text according to their specific requirements. To facilitate further research in visual text generation, we construct a training benchmark dataset called LAION-Glyph. We evaluate the effectiveness of our approach by measuring OCR-based metrics, CLIP score, and FID of the generated visual text. Our empirical evaluations demonstrate that GlyphControl outperforms the recent DeepFloyd IF approach in terms of OCR accuracy, CLIP score, and FID, highlighting the efficacy of our method.

# 1 Introduction

Denoising diffusion probabilistic models [34, 5, 35, 29, 31, 30, 16] have significantly boosted the development of general text-to-image generation by showing the capability of generating surprisingly high-quality images over the past few years. Although currently plenty of diffusion-based text-to-image generation methods could produce abundant fantastic and photo-realistic images, most existing methods still lack the ability to produce legible and readable text in generated images [29, 30] due to the complex and fine-grained structure within the visual text.

Several very recent efforts have made preliminary attempts to address the task of visual text generation. Motivated by the inspiring analysis in unCLIP [29], the spelling information inside the prompts can not be accurately modeled with the raw CLIP text embedding, the follow-up efforts including eDiff-I [3] and Imagen [31] attempt to leverage the potential of large language models such as T5 [28], which is trained on the text-only corpus, as the text encoder in image generation. With the strength of T5 embedding on encoding individual objects within the prompts [3], eDiff-I produces more accurate visual text. The very recent DeepFloyd IF model further follows the design of Imagen and demonstrates impressive performance in rendering legible text. Besides, [20] found that the text encoders (both CLIP and T5) used in most existing mainstream text-to-image generation models lack sufficient character-level information for spelling due to the usage of BPE tokenizer, thus they verify that adopting the character-aware model ByT5 [36] instead could bring significant improvements.

Despite efforts to modify text encoders used in generation models, layout errors such as missing or merged glyphs still exist in generated images [20], implying that merely relying on textual input prompts would not be sufficient for accurate visual text rendering. To address this problem, we propose to incorporate text glyph information into the off-the-shelf powerful text-to-image generation models for visual text generation. We formulate the task of visual text generation as a glyph-conditional control problem. Specifically, we propose to control the visual text generation with an additional glyph image.[3] The glyph image acts as an explicit spatial layout prior to enforcing the diffusion models generating coherent and well-formed visual text. We show some qualitative results in Figure 1 to show that our method is capable of generating diverse images with well-formed visual text.

In our implementation, we introduce two key innovations including (i) a GlyphControl framework that can augment the off-the-shelf text-to-image generation model by exploiting the shape information encoded in the glyph image with a ControlNet branch, and (ii) a LAION-Glyph benchmark that consists of $\sim$ 10 M text-image pairs augmented with additional OCR detection results that record the presented text information. We further create two evaluation benchmarks, including SimpleBench and CreativeBench, to assess the performance of our method and the other strong methods such as DeepFloyd IF. To demonstrate the effectiveness of our approach, we conduct thorough experiments and show that our approach consistently achieves much higher OCR accuracy than the DeepFloyd IF. For example, on SimpleBench and CreativeBench, our approach gains +15% (48% vs. 33%) and +13% (34% vs. 21%) than the very recent powerful DeepFloyd (IF-I-XL) while only requiring less than 22% parameters. We summarize our main contributions as follows:

---

[3]The glyph image is a whiteboard image where the characters are rendered with a single particular font while keeping the same content, position, and size as the realistic visual text.

- We propose a glyph-conditional text-to-image generation model named GlyphControl for visual text generation, which outperforms DeepFloyd IF, SDXL, and Stable Diffusion in terms of OCR accuracy, CLIP score, and FID while saving the number of parameters by more than $3\times$ compared to DeepFloyd IF-I-XL.

- We introduce a visual text generation benchmark named LAION-Glyph by filtering the LAION-2B-en and selecting the images with rich visual text content by using the modern OCR system. We conduct experiments on three different dataset scales: LAION-Glyph-100K, LAION-Glyph-1M, and LAION-Glyph-10M.

- We report flexible and customized visual text generation results. We empirically show that the users can control the content, locations, and sizes of generated visual text through the interface of glyph instructions.

## 2 Related Work

**Text-to-image Diffusion Models.** Denoising Diffusion Probabilistic Model [11] and its successors [24, 29, 3, 30, 31, 27] have demonstrated impressive performance on high-quality image synthesis with text prompts. GLIDE [24] emphasizes the necessity of the classifier-free guidance over CLIP guidance and the usage of cascaded diffusion models [12, 31, 3, 29] for high-fidelity, high-resolution generation. Imagen [31] introduces generic large language models (T5-XXL text encoder) into the text-to-image generation while demonstrating comparable or superior image quality to the CLIP text encoder. Moreover, eDiff-I [3] concatenates the CLIP text embeddings and T5 text embeddings to benefit from the strengths of both two text encoders. Unlike the aforementioned pixel-level diffusion models, Latent Diffusion [30] transforms the image into latent features and applies the diffusion model in the latent space to decrease training and inference costs. Stable Diffusion is an application of the Latent Diffusion method in text-to-image generation but trained with additional data and a powerful CLIP text encoder. SDXL[27] improves Stable Diffusion by using $3\times$ larger U-Net and a second text encoder and also introducing another refinement model to further enhance image quality through image-to-image techniques. In this work, we adopt Stable Diffusion as the base model.

**Controllable Image Generation.** To achieve more customized image synthesis, users could apply additional conditions, such as segmentation maps or depth maps [30], onto diffusion models. Beyond this intuitive approach, multiple diffusion-based methods of image editing [22, 15, 25, 8] demonstrate promising performance in controlling the content of synthesized images. Recently, more related works [37, 23, 14] have focused on flexible and composable control of image synthesis. Composer [14] decomposes the image generation into multiple factors and generates images by re-combining them. Both T2IAdapter [23] and ControlNet [37] can incorporate different conditional maps, such as segmentation maps or depth maps, as additional data into the pre-trained diffusion models, demonstrating accurate structure or color control without dampening the generation ability of the original models. Considering the fact that the glyphs of visual text essentially belong to geometric structures, we adopt ControlNet as the basic framework to generate visual text by controlling the local structure with additional glyphs.

**Visual Text Generation.** Although diffusion models could generate high-fidelity images, current mainstream text-to-image generation models such as unCLIP [29] and Stable Diffusion have trouble rendering legible and readable text onto images. Several previous works [9, 10] demonstrate that diffusion models have the ability to generate visual text with different fonts but do not extend to general image generation. Due to the findings that CLIP embedding could not precisely perceive the spelling information in the input prompts [29, 3], both Imagen [31] and eDiff-I [3] utilize the large language model T5 [28] to achieve superior visual text generation. The recent open-sourced image generation model DeepFloyd IF [16], inspired by Imagen, takes the T5-XXL as the text encoder as well demonstrating impressive performance in visual text generation. Furthermore, [20] thoroughly exploits the strengths of character-aware language models like ByT5 [36] over character-blind counterparts such as mainly used CLIP and T5. With the usage of ByT5 in the generation, the semantic errors of rendered text decrease while the errors related to the layout of glyphs still exist, implying that the auxiliary information about glyph images would be necessary. The very recent DALL·E 3 [26, 4] not only excels in rendering precise visual text but also showcases exceptional typography quality. GlyphDraw [21] successfully renders Chinese characters onto images by adding glyph images into the input of the diffusion model and also fusing extracted glyph embedding with

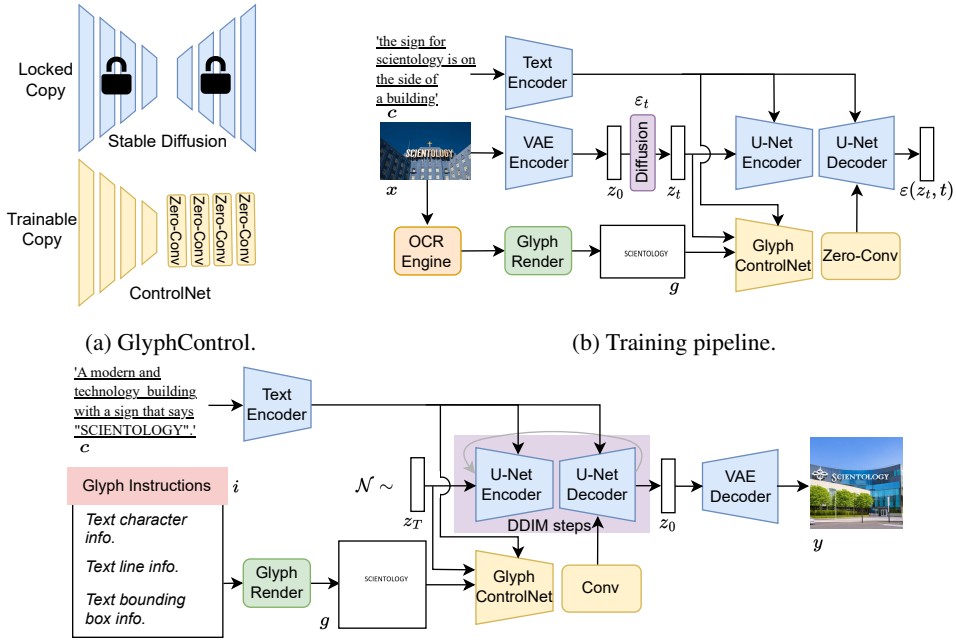

(a) GlyphControl.

(b) Training pipeline.

(c) Inference pipeline.

Figure 2: **Illustrating the framework of GlyphControl.** (a) The GlyphControl architecture comprises a pre-trained Stable Diffusion model as a "locked copy" and a randomly initialized ControlNet model as a "trainable copy." (b) During the training process, the input image $x$ undergoes encoding with a VAE encoder, resulting in a latent embedding $z_0$. The diffusion process is then applied to $z_0$, generating a noised latent embedding $z_t$. Additionally, we utilize an OCR engine (PP-OCR [7]) to extract text from images and employ a glyph render to generate a whiteboard image. This image exclusively represents recognized characters as black regions, forming the glyph image $g$. Consequently, both the text embedding (based on text caption $c$) and the noised latent embedding are fed into the U-Net (locked copy) and the Glyph ControlNet (trainable copy). This enables the estimation of the noise term $\varepsilon(z_t, t)$, with the crucial step involving passing the glyph image to the Glyph ControlNet to extract vital glyph information for rendering well-formed text. (c) During inference, our method supports diverse user instructions for customizing the rendering of the glyph image $g$. Subsequently, we sample a noise latent embedding $z_T$ from Gaussian noise and employ the DDIM scheme to perform the denoising process, estimating the denoised latent embedding $z_0$. Finally, $z_0$ is sent to the VAE decoder, resulting in the construction of the final output image $y$.

text embedding as a condition. Based on similar insights, we utilize glyph images as conditional maps to control image synthesis. Compared to the above methods, we could specify the contents, locations, and sizes of text, which brings more customized and flexible designs.

## 3  Approach

### 3.1  Preliminary

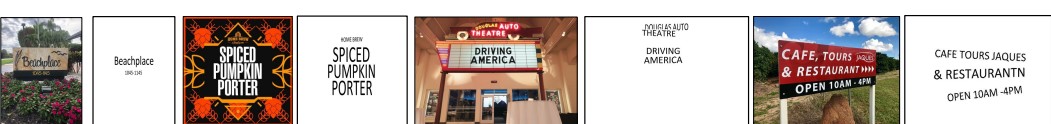

Figure 3: Illustrating of the generated glyph images based on the glyph render (LAION-Glyph-1M).

**Stable Diffusion [30].**   We have selected the "stable-diffusion-2-base" (SD 2.0-base[4]) as the foundational model in this work. The Stable Diffusion model is a highly capable and versatile text-to-image generative model that has been meticulously trained from scratch. The training process of basic

---

[4]https://huggingface.co/stabilityai/stable-diffusion-2-base

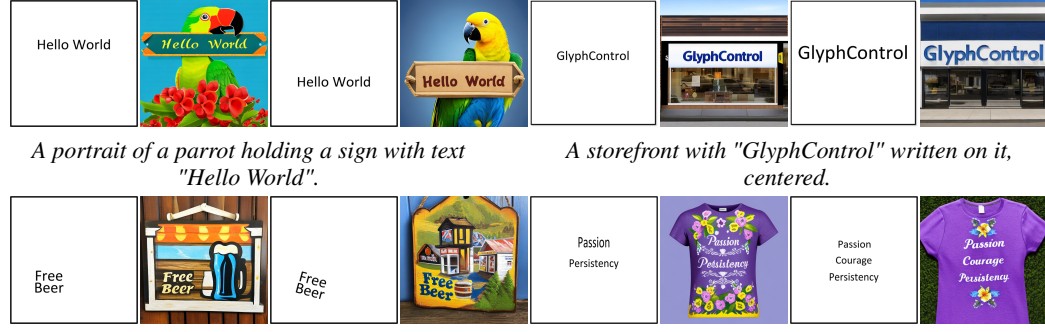

*A portrait of a parrot holding a sign with text "Hello World".*

*A storefront with "GlyphControl" written on it, centered.*

*A hand-painted wooden "Free Beer" sign hanging out of a bar.*

*A fancy violet T-shirt decorated with flowers while the message [X] are written on it.*

Figure 4: Illustrating the qualitative results of GlyphControl in terms of flexible user controllability. The whiteboard images depict the corresponding glyph condition maps alongside the generated images on the right side. The "[X]" symbol (in the last example) is used as a replacement for the rendered words on the T-shirt.

models involves 550k steps at resolution $256 \times 256$, focusing on a subset, with an aesthetic score of 4.5 or higher, of the LAION-5B dataset. What makes the difference between stable-diffusion-2-base and previous versions is that the model is continuously trained on the same dataset with a resolution of at least $512 \times 512$ pixels, which contributes to the model's ability to generate more detailed and visually appealing images. The training of stable-diffusion-2-base costs hundreds of hours with $128\times$ A100 GPUs. In this work, by employing the off-the-shelf "stable-diffusion-2-base" model and refining it through rigorous training processes, we aim to achieve superior results in the visual text generation domain.

**ControlNet [37].** The ControlNet is a powerful network that enhances pre-trained large diffusion models like Stable Diffusion with additional input conditions. It learns task-specific conditions in an end-to-end manner, even with limited training data. The network has a trainable copy and a locked copy of the diffusion model's weights, enabling it to retain general capabilities while fine-tuning for specific tasks. The ControlNet incorporates "zero convolution" layers to connect the trainable and locked components, gradually adapting convolution weights from zeros to optimized parameters. This allows precise control over the model's behavior in various applications.

### 3.2   GlyphControl

**Framework.**   The GlyphControl framework consists of several key components: (i) an OCR engine for detecting text information in the given image, (ii) a Glyph render for rendering the detected text in a whiteboard image at corresponding locations, (iii) an image VAE encoder that projects the input image into a latent code, and an image VAE decoder that reconstructs an output image based on the latent code, (iv) a text encoder (OpenAI CLIP text encoder) that converts the input text into text embedding, (v) a U-Net encoder and decoder that performs the denoising diffusion process, and (vi) a Glyph ControlNet that encodes the conditional glyph information by processing the glyph image rendered by the Glyph render. More details of the GlyphControl framework can be seen in Figure 2. Furthermore, Figure 3 showcases some example images of the rendered glyph images.

To incorporate glyph information, we introduce the concept of glyph input conditions by rendering glyph images and feeding them into the ControlNet branch. Unlike conventional conditions used in the original ControlNet [37], accurate visual text rendering greatly benefits from the use of rendered glyph images. We specifically chose the ControlNet architecture for its proficiency in controlling precise geometric structures.

With our GlyphControl approach, we can successfully generate legible and readable visual text. This is achieved by utilizing pre-rendered glyph images as input condition maps for the ControlNet, allowing us to control the generated glyphs at the layout level. Furthermore, we specify the words in the input text prompts (e.g., "A storefront with "GlyphControl" written on it") and leverage the CLIP text encoder to understand the semantic meaning of the words.

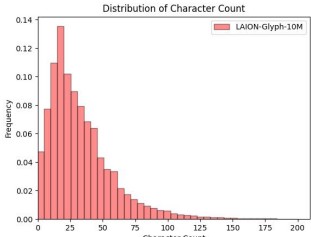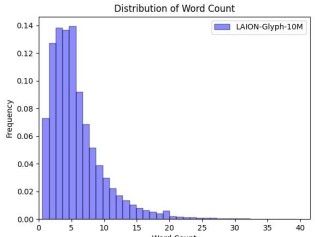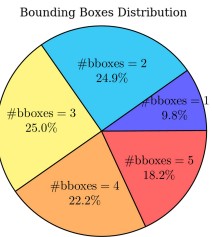

Figure 5: Statistics on LAION-Glyph-10M. Left: Distribution of character counts in each image. Middle: Distribution of word counts in each image. Right: Distribution of detected bounding boxes in each image.

**Glyph Instructions.**    One major advantage of our GlyphControl approach is its ability to support customized glyph instructions ($i$), which enables the specification of various constraints on the rendered text in the final output image. Our GlyphControl framework provides support for three types of text information customization:

- ■ **Text character information**: GlyphControl allows for the specification of not only single words but also phrases or sentences composed of multiple words. As long as the text is intended to be placed within the same area, users can customize the text accordingly.

- ■ **Text line information**: GlyphControl provides the flexibility to assign words to multiple lines by adjusting the number of rows. This feature enhances the visual effects and allows for more versatile text arrangements.

- ■ **Text box information**: With GlyphControl, users have control over the font size of the rendered text by modifying the *width* property of the text bounding box. The location of the text on the image can be specified using the *coordinates* property of the top left corner. Additionally, the *yaw rotation angle* property of the text box allows for further adjustments. By default, the text is rendered following the optimal width-height ratio, but users can define a specific *width-height ratio* to precisely control the height of the text box.

We demonstrate the effectiveness of these glyph instructions in Figure 4, where our approach successfully generates legible text according to the specified instructions. For instance, in Figure 4, we showcase examples where users can customize the positions of the rendered text, adjust the font size, or place multiple groups of text at different locations to achieve personalized designs. Additionally, users have the option to split the text into multiple rows or rotate the text boxes for improved arrangement. Our controllable text generation approach opens up possibilities for automated personalized art designs in the future. Moreover, in the experimental section, we provide empirical evidence showcasing that our method achieves significantly higher OCR accuracy compared to the recent DeepFloyd model.

**Implementation.**    We utilize the same architecture and initial weights for the VAE and U-Net as the SD 2.0-base model and adopt the classifier-free guidance[13] as the previous works do. Our training process incorporates PP-OCRv3 [6] as the OCR engine. During inference, users need to provide glyph instructions to generate customized images. For rendering glyphs, we leverage the tools available in the ImageDraw module of the Python library Pillow.

### 3.3   LAION-Glyph Benchmark

**Overview.**    The training process of Stable Diffusion and DeepFloyd has greatly benefited from the utilization of the extensive multi-modal dataset LAION-5B [32]. However, there is currently a notable absence of openly available datasets specifically tailored for visual text generation tasks. To bridge this gap, we introduce the LAION-Glyph benchmark. To construct this benchmark, we start with LAION-2B-en, which is a subset of LAION-5B [32], and selectively choose specimens that exhibit abundant visual text content using the PP-OCR engine.

**Pipeline.**    Our data construction process consists of two consecutive steps. In the first step, we apply an aesthetic score prediction model to filter out images with an aesthetic score higher than 4.5. Next, we utilize the PP-OCRv3 [6] engine for text detection and recognition. To ensure the quality

| Method | #Params | Text Encoder | Training Dataset | Acc(%)↑ | A$\hat{}$cc(%)↑ | LD ↓ |
|---|---|---|---|---|---|---|
| Stable Diffusion v2.0 | 865M | CLIP(354M) | LAION 1.2B | 0/0 | 3/2 | 4.25/5.01 |
| SDXL 1.0 | 5.8B | CLIP & OpenCLIP(817M) | Internal Dataset (>100M) | 0.3/0.5 | 13/8 | 6.26/6.30 |
| DeepFloyd (IF-I-M) | 2.1B | T5-XXL(4.8B) | LAION 1.2B | 0.3/0.1 | 18/11 | 2.44/3.86 |
| DeepFloyd (IF-I-L) | 2.6B | T5-XXL(4.8B) | LAION 1.2B | 0.3/0.7 | 26/17 | 1.97/3.37 |
| DeepFloyd (IF-I-XL) | 6.0B | T5-XXL(4.8B) | LAION 1.2B | 0.6/1 | 33/21 | 1.63/3.09 |
| GlyphControl | 1.3B | CLIP(354M) | LAION-Glyph-100K | 30/19 | 37/24 | 1.77/2.58 |
| GlyphControl | 1.3B | CLIP(354M) | LAION-Glyph-1M | 40/26 | 45/30 | 1.59/2.47 |
| GlyphControl | 1.3B | CLIP(354M) | LAION-Glyph-10M | **42/28** | **48/34** | **1.43/2.40** |

Table 1: Comparison results of OCR-related metrics with prior methods in the field of visual text generation is shown in the table. The results are averaged over four word-frequency buckets. The results on SimpleBench/CreativeBench are presented on the left/right side of the slash, respectively. It is important to note that the total number of parameters reported in the second column of the table does not include the text encoder. The LAION 1.2B dataset comprises image-text pairs with predicted aesthetic scores of 4.5 or higher in LAION 5B. All the DeepFloyd models use IF-II-L (1.2B) and Stable ×4 as the upscale models to progressively increase the image resolutions from $64 \times 64$ to $1024 \times 1024$. SDXL generates images with a resolution of $1024 \times 1024$.

of the data, we discard images where all OCR boxes are located at the image border. Additionally, we remove images that have total OCR areas less than 5% of the whole image area or contain more than 5 bounding boxes, as these cases may lead to text recognition or image reconstruction failures. To address inaccuracies in the original captions from the LAION dataset, we generate new captions using the BLIP-2 [17] model. As a result, we have curated a high-quality LAION-Glyph dataset consisting of 10 million images. This dataset includes detailed OCR information and captions that are well-formed and accurate.

**Statistics.** As illustrated in Figure 5, the character count in the images is primarily concentrated within the range of 10 to 50 characters, with the majority of samples containing fewer than 150 characters. In terms of word distribution, the most common cases consist of 3 to 5 words, while instances with more than 15 words are relatively rare. Additionally, the number of bounding boxes is fairly evenly distributed, although images with only one box are less prevalent. To facilitate training and evaluation, we partitioned the LAION-Glyph dataset into three scales: LAION-Glyph-100K, LAION-Glyph-1M, and LAION-Glyph-10M, using a random division approach.

## 4 Experiment

### 4.1 Training Details

We train our framework on three different dataset scales: LAION-Glyph-100K, LAION-Glyph-1M, and LAION-Glyph-10M for $60\times$ epochs, $20\times$ epochs, and $6\times$ epochs, respectively. The initial weights of both the SD branch and Glyph ControlNet branch are copied from the SD 2.0-base model. For both the Glyph ControlNet and Zero-Conv blocks, we set the base learning rate to $1e-4$. The U-Net encoder and decoder are both kept frozen during training. The caption dropping rates for the SD branch and Glyph ControlNet branch are set to $0.1$ and $0.5$, respectively. The input images are maintained at a resolution of $512 \times 512$.

### 4.2 Evaluation

**Metrics.** We evaluate the effect of visual text generation on OCR accuracy. We measure OCR exact match accuracy, denoted as **Acc**, which assesses the word-level agreement between the OCR recognition results and the ground truth visual text. In other words, it represents the complement of the Word Error Rate (WER), i.e., $1-$WER. As the DeepFloyd model tends to generate visual text in all capital letters, regardless of the original form of the words in the prompt, we introduce the OCR capitalization-insensitive exact match accuracy **A$\hat{}$cc**. This measure allows for a fairer comparison by disregarding the case of the text. Additionally, we incorporate character-level OCR accuracy by using the Levenshtein distance for partial matching evaluation. We report the average Levenshtein distance **LD** for each word, providing insights into the accuracy at the character level.

In addition to the OCR-based metrics mentioned earlier, we evaluate the image-text alignment of the generated visual text images using the CLIP score, as done in previous works [3, 31]. To assess the

| Method | Stable Diffusion v2.0 | SDXL 1.0 | DeepFloyd (IF-I-M) | DeepFloyd (IF-I-L) | DeepFloyd (IF-I-XL) | GlyphControl-100K | GlyphControl-1M | GlyphControl-10M |
|---|---|---|---|---|---|---|---|---|
| CLIP Score↑ | 31.6/33.8 | 31.9/33.3 | 32.8/34.3 | 33.1/34.9 | 33.5/35.2 | 33.7/36.2 | 33.4/36.0 | **33.9/36.2** |
| FID-10K-LAION-Glyph↓ | 34.03 | 44.77 | 23.37 | 30.97 | 26.58 | **22.04** | 22.19 | 22.22 |

Table 2: Illustrating the CLIP score and FID comparison results based on the settings described in Table 1. The average results of CLIP scores across four word frequency buckets on both benchmarks are provided.

quality of generated images, we calculate FID scores using 10K samples, which have not been used for training in our LAION-Glyph dataset.

**Benchmark.** We construct two evaluation benchmarks by incorporating prompt templates from previous works on visual text generation [20, 21] and embedding different words selected by us into these templates.

- **SimpleBench**: A simple text prompt benchmark following [20]. The format of prompts remains the same: *'A sign that says "<word>".'*
- **CreativeBench**: A creative text prompt benchmark adapted from GlyphDraw [21]. We adopt diverse English-version prompts in the original benchmark and replace the words inside quotes. As an example, the prompt may look like: *'Little panda holding a sign that says "<word>".'* or *'A photographer wears a t-shirt with the word "<word>" printed on it.'*

In accordance with [20], we collect a pool of single-word candidates from Wikipedia. These words are then categorized into four buckets based on their frequencies: $\text{Bucket}_{\text{top}}^{\text{1k}}$, $\text{Bucket}_{\text{1k}}^{\text{10k}}$, $\text{Bucket}_{\text{10k}}^{\text{100k}}$, and $\text{Bucket}_{\text{100k}}^{\text{plus}}$. Each bucket contains words with frequencies in the respective range. To form input prompts, we randomly select 100 words from each bucket and insert them into the aforementioned templates. Consequently, we generate four images for each word during the evaluation process.

**Inference Details.** The scale of classifier-free guidance is set as 9 while we take the empty string as the negative prompt. We use the DDIM[35] sampler with 20 sampling steps.

## 4.3 Main Results

Table 1 presents the comparison of our method with the most representative generative models, including Stable Diffusion, SDXL, and DeepFloyd. Due to a lack of fully open-sourced codes and checkpoints, we could not conduct fair quantitative comparisons with previous works on visual text generation [21, 20]. Our method achieves the highest OCR accuracy on both benchmarks compared to other methods. Notably, the OCR accuracy of Stable Diffusion is almost zero, indicating that it is unable to generate legible text. While for SDXL, an improved version of Stable Diffusion, although the case-insensitive OCR accuracy is improved due to adopting larger text encoders, the overall OCR performance is still poor. In contrast, our framework, with the addition of glyph control, enables the diffusion models to render relatively accurate visual text while maintaining fewer training parameters to the original Stable Diffusion model. Compared to multiple versions of the pixel-level diffusion model DeepFloyd IF with T5 text encoder, our method could achieve better OCR performance with fewer parameters. Additionally, other compared methods tend to generate capital characters regardless of the original word form, resulting in much lower performance on case-sensitive metrics and reduced flexibility in real-world usage.

The comparison among the DeepFloyd IF models demonstrates that increasing the model parameters can enhance the accuracy of generated visual text. Similarly, for our method, training on a larger dataset can improve text generation performance. For example, the OCR accuracy $\hat{\text{Acc}}$ on SimpleBench increased from $37\%$ to $48\%$ when trained on a larger specialized visual text-related dataset. This highlights the importance of leveraging larger datasets specifically focused on visual text in order to achieve further improvements in performance.

In addition to the OCR-related evaluation, we also assess the consistency between prompts and generated images, as shown in Table 2. Our method outperforms other general text-to-image models or achieves comparable results in terms of the CLIP scores. This indicates that our model has the capability to accurately render the specified words in the text prompts within the generated images, while still maintaining the fundamental ability to align image and text. While for FID scores

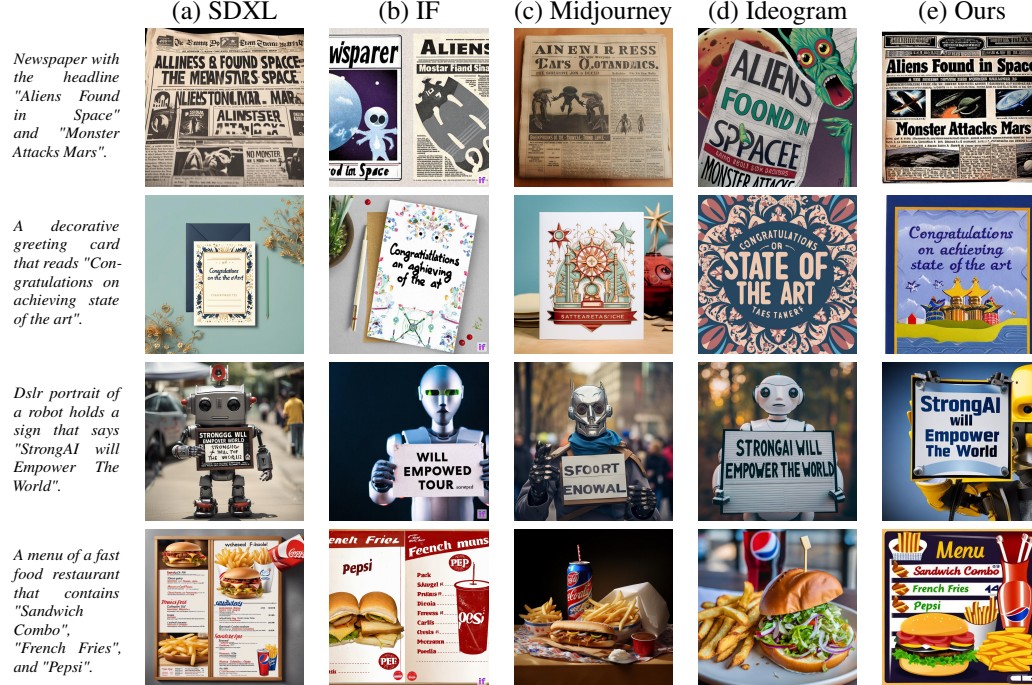

|  | (a) SDXL | (b) IF | (c) Midjourney | (d) Ideogram | (e) Ours |
|---|---|---|---|---|---|

*Newspaper with the headline "Aliens Found in Space" and "Monster Attacks Mars".*

*A decorative greeting card that reads "Congratulations on achieving state of the art".*

*Dslr portrait of a robot holds a sign that says "StrongAI will Empower The World".*

*A menu of a fast food restaurant that contains "Sandwich Combo", "French Fries", and "Pepsi".*

Figure 6: Qualitative comparison results. The left column presents the text prompt and the other five columns show the images generated by Stable Diffusion XL 1.0 (SDXL), DeepFloyd IF-I-XL (IF), Midjourney, Ideogram AI, and our GlyphControl. The results demonstrate that competitive baseline methods exhibit limitations in generating text, including typos and incomplete text generation.

evaluated on the LAION-Glyph dataset, our method achieves the lowest FID score compared to other text-to-image models, implying that GlyphControl could generate realistic visual text images with high quality. Furthermore, we conduct a comparison of the generation performance across different benchmarks and word buckets (see Figure 10).

## 4.4 Qualitative Analysis

Through visualization of generated images (shown in Figure 8 & 6), we compare our GlyphControl with both competitive open-sourced text-to-image generation methods (SD, SDXL, and IF) and some leading billing software (DALL·E 2[29], Midjourney [2], and Ideogram [1]).

As depicted in Figure 8, Stable Diffusion, DALL·E 2, SDXL, and DeepFloyd IF exhibit various types of errors during text rendering, including missing glyphs (the 1st sample), repeated or merged glyphs (the 3rd & 6th samples), and misshapen or wrong glyphs (the 5th & 7th samples). In some challenging cases (the 2nd & 4th samples), these models even fail to generate any text at all. In contrast, our method, which incorporates additional glyph images as structure control, could accurately generate legible text that aligns well with the input text prompts by providing appropriate glyph instructions.

We also employ other compared methods to generate images with the same text prompts in Figure 1 (Figure 6). For the cases of rendering longer phrases or sentences (the 2nd & 3rd samples), SDXL, IF, and Midjourney fail to generate complete readable visual text while Ideogram, a recently published AI-based tool specializing in generating visual text, could generate legible text with fewer errors. By comparison, our method outperforms other methods in terms of the accuracy of text rendering. Furthermore, although current top text-to-image tools like Midjourney could generate high-fidelity images, these models including Ideogram still struggle to handle the cases where multiple groups of visual text need to be rendered at various locations within the images (the 1st & 4th samples of Figure 6). While our method could effectively translate such combinations of visual text.

## 4.5 Ablation Experiments

**Ablation on Font Size.** GlyphControl supports users to control the font size of the rendered text by modifying the width property of the text bounding box. We further report the generation results of

| Font Size | Acc(%)↑ | Âcc(%)↑ | LD ↓ | CLIP Score↑ |
|-----------|---------|---------|------|-------------|
| Small | 5/4 | 10/7 | 4.85/5.51 | 31.7/33.4 |
| Medium | **30** / 19 | **37/24** | **1.77** / 2.58 | **33.7/36.2** |
| Large | 23 / **20** | 27/23 | 1.94 / **2.37** | 33.1/35.7 |

Table 3: Ablations on font sizes. The font size of the rendered text refers to the *width* of the text bounding box. We adopt the model trained on LAION-Glyph-100K for ablations. The results under the "Medium" setting are exactly the same as those in Table 1 & 2.

various font sizes using GlyphControl (Table 3). For small font sizes, both the OCR accuracy and CLIP score of the generated images substantially decrease, which could be attributed to the relatively smaller area of rendered glyphs within the entire image. While for large font sizes, the generation performance becomes worse probably because of the limited number of such training samples containing large glyphs. In order to attain better generation performance, choosing appropriate font sizes or other glyph instructions would be critical.

**Ablation on a Large Amount of Small Text.** We further study the scenarios of generating paragraphs, i.e., numerous groups (OCR boxes) of small text, rather than simple phrases or words. As Figure 9 shows, our method could preserve the global arrangement following glyph instructions, but fail to generate readable small-size text within paragraphs. We also conduct an analysis of other failure cases shown in Appendix C.

## 5 Discussion and Limitations

While our method offers flexible control over the positions and sizes of rendered text using glyph instructions, it currently lacks the ability to control font style and text color. We only render black text in a single default font style onto whiteboard conditional maps for both training and inference. To enhance our approach, we aim to explore advanced font rendering integration, such as artistic fonts, with GlyphControl and investigate modern font style recognizers to extract style information from training data, incorporating it into text captions during training. For controlling font color, incorporating a color spatial palette adapter[23] into the framework may prove effective.

As illustrated in the ablation experiments in Section 4.5, our method exhibits sub-optimal performance when generating a large amount of text with small font sizes. This issue could be attributed to the removal of samples containing more than five OCR boxes during training. In future work, we aim to relax these restrictions on the filtering of the LAION-Glyph dataset, increasing the threshold from five to ten or twenty bounding boxes. It is worth noting that rendering small text presents challenges for VAE-based pixel-level reconstruction due to the text's relatively diminutive size and the dense distribution of text, which may affect both OCR recognition and glyph rendering. Additionally, we aspire to apply the GlyphControl framework to high-resolution text-to-image methods, such as SDXL at $1024 \times 1024$, and explore further possibilities for local editing of visual text within generated images. Last, we observed that BLIP-2 captions often encounter difficulties in consistently representing both image content and OCR text information. Investigating more powerful caption models, such as those presented in [19, 18], is a worthwhile endeavor.

## 6 Conclusion

In this paper, we introduced the GlyphControl method, a remarkably simple yet highly effective approach for generating legible and well-formed visual text. The success of our approach is attributed to two key contributions: (i) the employment of the glyph ControlNet, which encodes text shape information based on rendered glyph images, and (ii) the establishment of the LAION-Glyph benchmark, benefiting from large-scale training data. By integrating the GlyphControl method with the LAION-Glyph benchmark, our approach achieves exceptional performance and consistently outperforms recent models such as the DeepFloyd IF in terms of OCR accuracy, FID, and CLIP score. Our work serves as a valuable foundation for future research in developing robust visual text generation models. We would like to further explore font style and color control, as well as address challenges related to generating abundant small text and improving the quality of captions.

**Acknowledgement.**    We are grateful to the anonymous reviewers for their invaluable feedback that greatly improved this work. We also thank Bohan Chen, Farzaneh Rajabi, Ji Li, and Chong Luo for their insightful suggestions post-submission.

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

## A  More Visualized Examples

We present additional samples generated by GlyphControl in Figure 7 as the supplementary of Figure 1, demonstrating the capability of GlyphControl to generate legible and well-formed visual text in diverse scenarios. Furthermore, Figure 8 provides more qualitative comparison results with competitive text-to-image models. Additionally, Figure 9 illustrates some GlyphControl-generated images that contain plentiful small text. For a detailed analysis of these results, please refer to Section 4.4 & 4.5 .

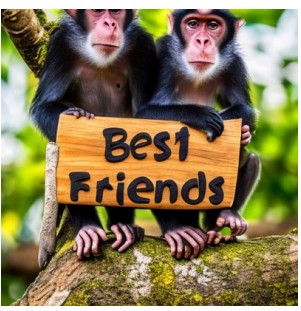

*A photo of two monkeys sitting on a tree. They are holding a wooden board that says "Best Friends", 4K dslr.*

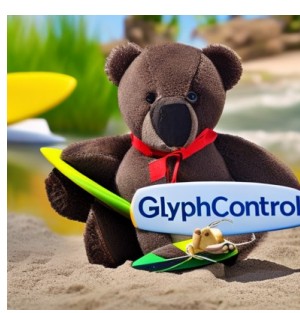

*A photo of a teddy bear holding a surfboard with the text "GlyphControl".*

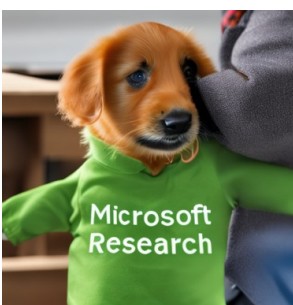

*A photo of a golden retriever puppy wearing a green shirt with text that says "Microsoft Research". Background office. 4k dslr*

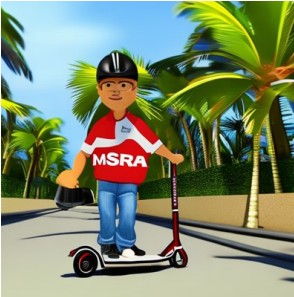

*A close-up 4k dslr photo of a cartoon man riding a scooter with text "MSRA". There are palm trees in the background.*

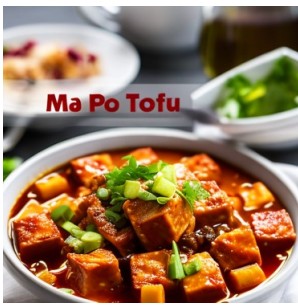

*A photo of a plate at a restaurant table with Ma Po Tofu and with text "Ma Po Tofu" on it. photorealistic, dslr.*

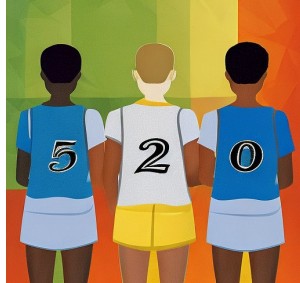

*Three people stand in a line and their backs with text "5", "2", and "0".*

Figure 7: More selected samples at $512 \times 512$ generated by GlyphControl using different text prompts and glyph conditions.

## B  Comparisons across Benchmarks and Word Frequency

In Figure 10, we demonstrate the OCR accuracy and CLIP scores based on different test benchmarks and word frequency buckets.

Generally, all methods exhibit higher OCR accuracy on the SimpleBench compared to the more challenging CreativeBench, which features more diverse prompts. However, CLIP scores tested on the SimpleBench are slightly lower than those on the CreativeBench, which could be attributed to the richer descriptions present in the prompts of the latter.

Additionally, as shown in in Figure 10, words with high-frequency appearances are generally easier to be rendered onto images compared to rarely-used words. An interesting finding is that low-frequency words consistently exhibit higher CLIP scores. This might be attributed to the CLIP embedding's tendency to overlook low-frequency words in the prompts, potentially leading to overestimation.

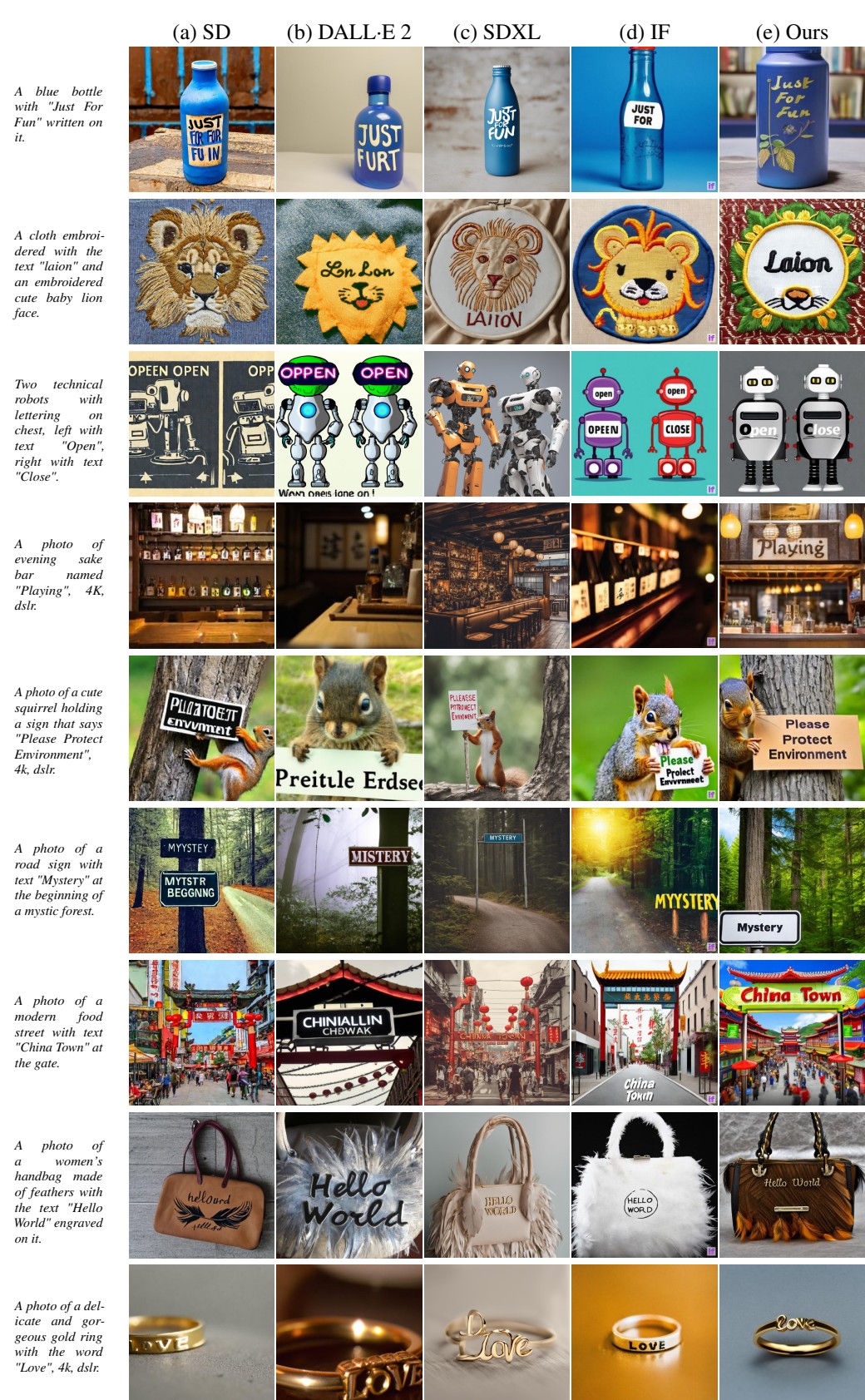

Figure 8: Qualitative comparison results among Stable Diffusion v2.0 (SD), DALL·E 2, Stable Diffusion XL 1.0 (SDXL), DeepFloyd IF-I-XL (IF), and our GlyphControl.

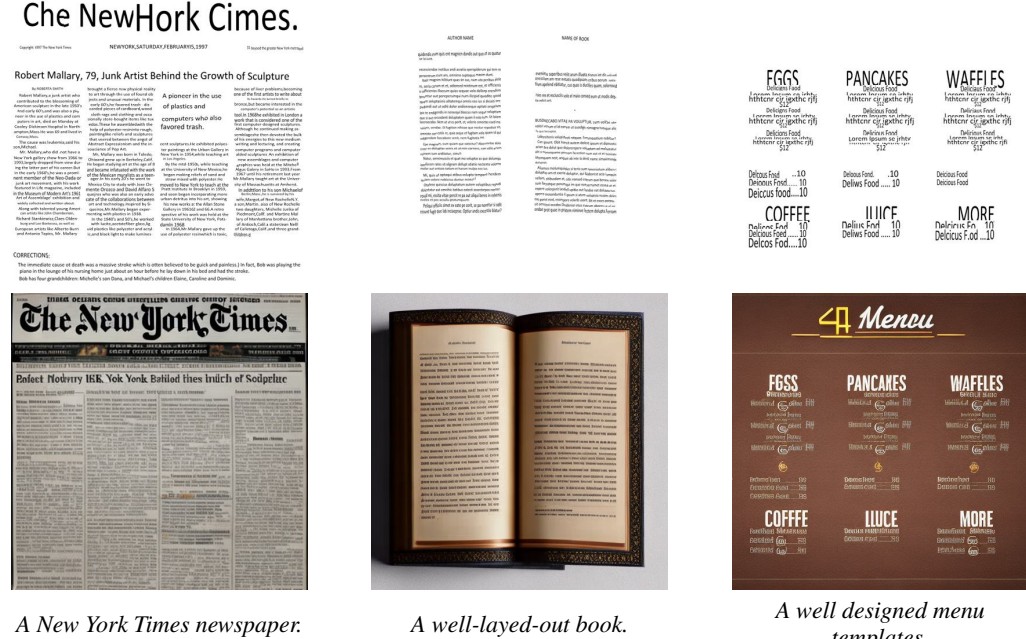

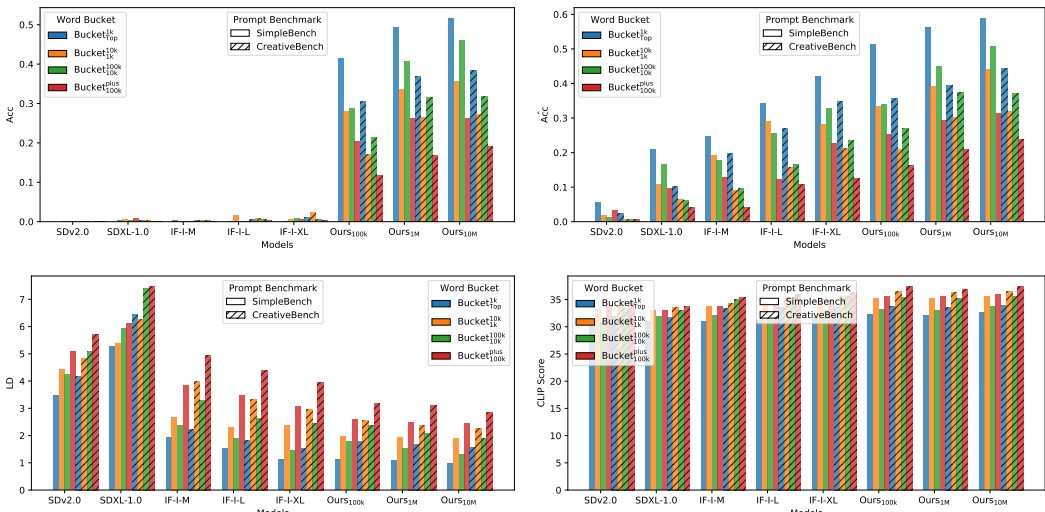

*A New York Times newspaper.*  *A well-layed-out book.*  *A well designed menu templates.*

Figure 9: Examples generated by GlyphControl in the scenario of a large amount of small text and OCR boxes. The results show that our method achieves relatively weak capability in generating abundant legible and readable small size text although preserving the global arrangement of glyph images.

Figure 10: Comparison of Stable Diffusion (SD), SDXL, DeepFloyd, and our GlyphControl across different benchmarks and frequency buckets.

## C  Failure Case Analysis

We show some failure cases generated by GlyphControl in Figure 11. The *Rendering Overlap* problem happens when the locations of text boxes within glyph instructions overlap. Although the text rendering accuracy has improved significantly compared to other text-to-image generation models (Section 4.3 & 4.4), some "layout issues"[20] like *Missing Glyphs*, *Wrong Glyphs*, *Duplicate Glyphs*, *Unexpected Text*, and *Illegible Text* still occur, implying the necessity of improvement of the generative model architecture or conditional control scheme. Bad performance in the scenarios of *Excessive Yaws* and *Small Text* may be attributed to a lack of corresponding training samples.

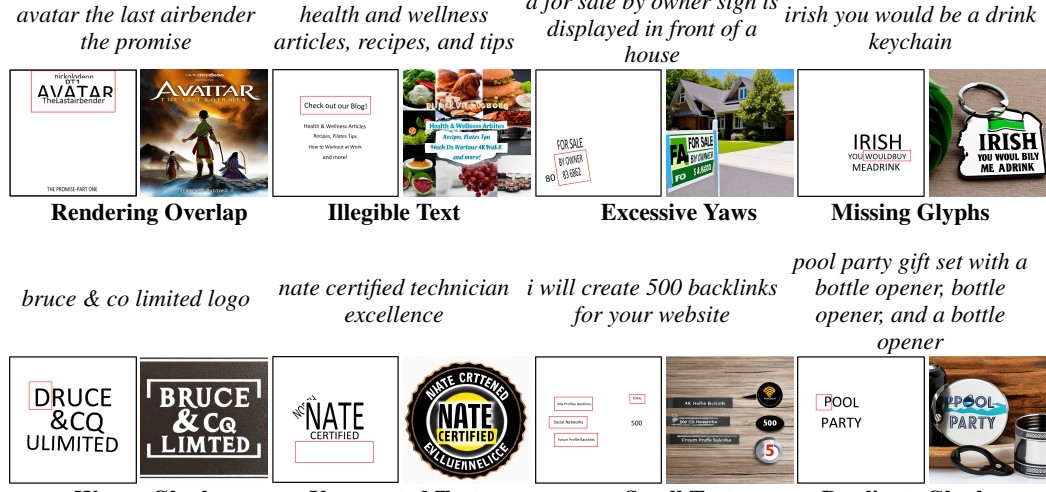

Figure 11: Failure cases generated by GlyphControl using captions and glyph instructions in the LAION-Glyph test dataset. We demonstrate 8 types of errors here. Captions and glyph images are also shown here. And we also mark out the places where the errors occur with red rectangles on the corresponding glyph images.

| Training Dataset | U-Net Decoder | **Acc**(%)↑ | **Âcc**(%)↑ | **LD** ↓ | CLIP Score↑ |
|---|---|---|---|---|---|
| TextCaps 5K | frozen | 48/39 | 55/43 | 1.21/1.73 | 33.8/36.3 |
| | fine-tuning | **61/43** | **68/49** | **0.76/1.38** | **34.2/36.3** |

(a) Illustrating the effect of unlocking the U-Net decoder: we conducted two experiments, in both cases, the models were fine-tuned for an additional 40 epochs using the checkpoint trained on LAION-Glyph-100K as the starting point (shown in Table 1). During the fine-tuning process, the learning rate of the decoder was set to 1e-4.

| Training Dataset | Training Epochs | **Acc**(%)↑ | **Âcc**(%)↑ | **LD** ↓ | CLIP Score↑ |
|---|---|---|---|---|---|
| TextCaps 5K | pre-trained | 30/19 | 37/24 | 1.77/2.58 | 33.7/36.2 |
| | 10 | 48/28 | 56/34 | 1.31/2.32 | 33.8/35.5 |
| | 20 | 50/34 | 61/41 | 1.03/2.01 | **34.3**/35.7 |
| | 40 | **61**/**43** | **68/49** | **0.76/1.38** | 34.2/**36.3** |

(b) The comparison between different training epochs. The U-Net decoder is also fine-tuned.

Table 4: Ablation experiments on TextCaps 5K. We report the average results on two benchmarks.

# D More Ablation Studies

To assess the generalization capability of our approach on different training datasets, we curate a specialized OCR-related training dataset called TextCaps 5K. This dataset consists of images related to signs, books, and posters, which are extracted from the training set of the TextCaps v0.1 Dataset [33]. The original purpose of this dataset was image captioning with an understanding of visual text content. We fine-tune our model on the TextCaps 5K dataset for additional 40 epochs, using the same training settings as those applied to the model trained on LAION-Glyph-100K (as shown in Table 1). This fine-tuning process aims to evaluate how well our model performs on a different training dataset and further validate its robustness and adaptability.

Through our experiments, we discover that unlocking the frozen U-Net decoder in the original Stable Diffusion model can significantly enhance the OCR accuracy of visual text rendering, as demonstrated in Table 4a. This improvement can be attributed to the decoder's improved adaptation to the smaller TextCaps 5K dataset during training. As training progresses, the accuracy of text generation gradually improves, as shown in Table 4b. However, it is worth noting that the generated images tend to resemble the samples in TextCaps 5K.

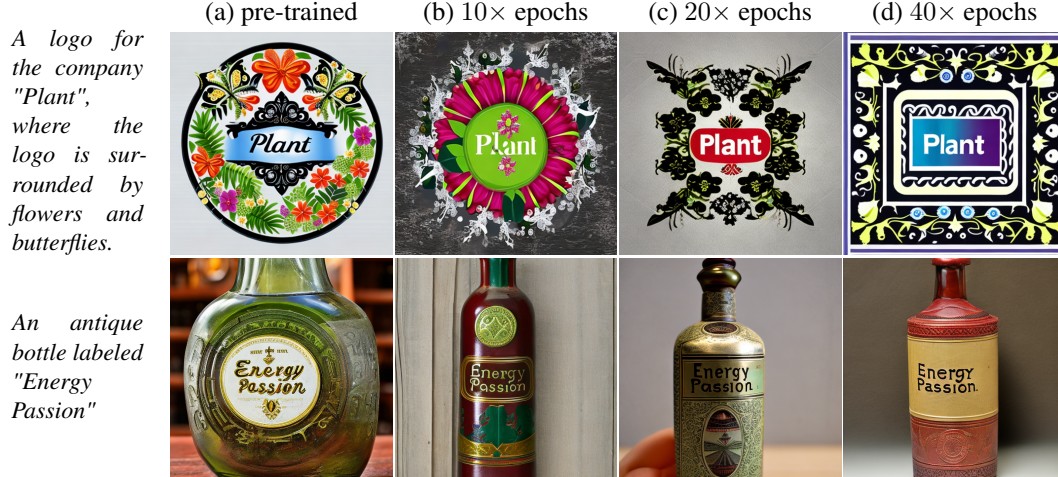

Figure 12: Illustrating the visual text images generated using the models fine-tuned on TextCaps 5K. The models were trained with the same settings following Table 4b.

As depicted in Figure 12, the visual text regions in the center of the generated images appear more like conventional signs (Figure 12d), without seamlessly merging with the background, while the images generated by the model pre-trained on LAION-Glyph-100K exhibit greater creativity and diversity (Figure 12a). Therefore, in order to generate photo-realistic images that accurately render visual text, it is essential to not only have a training dataset with a large number of high-quality samples containing abundant visual text content, but also to include more realistic images with diverse scenes and creative elements.

