# OpenReview forum: "GlyphControl: Glyph Conditional Control for Visual Text Generation"
_NeurIPS.cc/2023/Conference — NeurIPS 2023 poster_

### Official Review · Reviewer_Yxcw · 2023-07-03

**Soundness:** 4 excellent
**Presentation:** 4 excellent
**Contribution:** 3 good
**Rating:** 7
**Confidence:** 4

**Summary:**

This work proposes an approach to generating images with visual text. The main approach consists of a generalized version of ControlNet with rendered text as control guidance. To facilitate the training and evaluation, a benchmark dataset called LAION-OCR is proposed. Experiments show that the proposed approach outperforms competitors such as DeepFloyd IF and SD.

**Strengths:**

- Generating visual text accurately has been a challenging issue in the field. This work provides a simple yet effective approach to this challenge by elegantly extending ControlNet with rendered text as control. I am convinced that having rendered text as input makes a lot of sense in this context and would greatly help generate accurate text.
- In addition, having rendered text also allows control over the positioning, font size, which would be of great help in practice.
- Experiments show that the proposed approach significantly outperforms competitors while showing compelling visual results.
- Overall, I believe this work is a nice addition to the current research landscape of text-to-image generation.

**Weaknesses:**

It would help to add some failure cases analysis in order to better understand when models may fail.

**Questions:**

- Is the text encoder frozen? What about U-Net Encoder? What is the influence of such design choices?

**Limitations:**

Limitations not well discussed.

---

> ### Author Rebuttal · Authors · 2023-08-10
>
> ## Response to Reviewer Yxcw
>
> We thank the reviewer for the careful reviews and constructive suggestions. We answer the questions as follows.
>
> > "It would help to add some failure cases analysis in order to better understand when models may fail."
>
> A: Thanks for your suggestion. We have added additional failure cases of eight types of errors in **Figure 3 of the attached PDF file**. The Rendering Overlap problem happens when the locations of text boxes within glyph instructions overlap. Some layout issues like Missing Characters, Wrong Characters, and Duplicate Characters still occur. Bad performance in the scenarios of Excessive Yaws and Small Text may be attributed to lack of corresponding training samples.
>
> Besides, we also show some cases about generating a large amount of small font size text in **Figure 2 of the attached PDF file**. Although  the arrangement of glyph images is preserved within generated images, our model still struggles in generating readable small size text.
>
> And we will add the above failure case analysis to the revised paper.
>
> ---
>
> > "Is the text encoder frozen? What about U-Net Encoder? What is the influence of such design choices?"
>
> A: Yes, the text encoder CLIP is frozen. As for the U-Net Encoder, the part of the original Stable Diffusion is frozen while Glyph ControlNet (seen in Figure 2), which is essentially an additional copy of the U-Net Encoder, is trainable. Such design choices aim at preserving the original generation ability of SD.
>
> ---
>
> > "Limitations not well discussed."
>
> A: Thanks for your advice. Here are some limitations of our method:
> - Lack of controllability on text font, color, and style
> - Poor performance when generating abundant text or characters with small font sizes
> - Using samples with a limited number of OCR boxes for training
>
> We will include more discussions about the limitations and potential future work to our revised paper.

---

> > ### Comment · Reviewer_Yxcw · 2023-08-11
> >
> > The response addresses my questions. I believe this work provides an effective empirical solution to the target task. I'd therefore maintain my original rating.

---

> > > ### Author Response · Authors · 2023-08-11
> > > **Thanks for Reviewer Yxcw's Prompt Response**
> > >
> > > We appreciate the reviewer's thorough feedback and positive rating. We'll incorporate your valuable suggestions, including the rebuttal contents, in the final paper revision.

---

### Official Review · Reviewer_xxJ1 · 2023-07-07

**Soundness:** 3 good
**Presentation:** 3 good
**Contribution:** 3 good
**Rating:** 6
**Confidence:** 5

**Summary:**

This paper addresses the development of diffusion-based text-to-image generative models for generating coherent visual text. They propose GlyphControl, which augments textual prompts with glyph conditional information to encode shape details and improve accuracy. They introduce the LAION-OCR benchmark dataset and evaluate GlyphControl's effectiveness using OCR-based metrics and CLIP scores, demonstrating its superior performance over the DeepFloyd IF approach in empirical evaluations.

**Strengths:**

The strengths of this paper lie in the proposal of GlyphControl, a glyph-conditional text-to-image generation model. Additionally, the introduction of the LAION-OCR benchmark dataset and the ability to customize and control the content, locations, and sizes of generated visual text demonstrate the paper's practical contributions to the field of visual text generation.

**Weaknesses:**


When it comes to text generation, information such as font size and style is crucial, and it would be beneficial for the authors to conduct experimental analysis on this aspect.

The authors' decision to remove images with more than 5 bounding boxes lacks clarity. Considering rich-text images as a valuable corpus and only focusing on a limited number of OCR images may limit the model's ability to generate rich-text images.

While the authors utilize BLIP-2 captions, which may not consistently describe both the image content and OCR text information, it would be interesting to see how the authors address the challenge of generating captions that consider both aspects.

Although this method generates accurate text, it raises curiosity about its effectiveness when dealing with a large amount of text or small font sizes, such as generating a paragraph rather than a simple phrase or word.

In addition to text generation, is it possible to modify the text within an image based on given text information, such as changing the color and position of specific words?

When dealing with a large amount of text, such as 500 words, ensuring text generation quality becomes crucial, as it requires high-resolution images. Additionally, ensuring efficiency in generating such images is also an important consideration.

Many texts may not need to be generated as they can be obtained through text rendering. Have the authors attempted this approach, based on text rendering?

**Questions:**

See weakness part

**Limitations:**

The main concern of this paper revolves around the quantity of text. Starting from the database, the authors control the number of OCR boxes, which, to some extent, limits the generation of rich-text images.

---

> ### Author Rebuttal · Authors · 2023-08-10
>
> We thank the reviewer for reviews and suggestions.
>
> >  Q1
>
> A: Great point!
>
> 👉 For font sizes, our GlyphControl supports users to control the font size of the rendered text by modifying the width property of the text bounding box. We further report the generation results of various font sizes using GlyphControl-SDv2.0, trained with LAION-OCR-100K, in **Table 1 of the attached PDF**.
>
> 👉 As we have not included the font style during training, our method does not support controlling font style yet. We would like to investigate the modern font style recognizer to include the style information into the text caption during training.
>
> ---
>
> > Q2
>
> A: Great point!  We summarize the key reasons as follows:
>
> - First, we want to highlight that there exist more than 30% of images with more than 5 bounding boxes in our dataset. Here we remove them simply due to limited resources. We have further filtered out samples with the total OCR areas less than 5% of the whole image area, considering that these images do not have sufficient text information. Therefore, the remaining samples only comprise around 20% of the entire set with visual text. We believe there still exist lots of opportunities.
>
> - Second, it's pertinent to note that a majority of previous endeavors have predominantly showcased the capability to generate less than two bounding boxes. As a result, this simplifies the complexity of the text-rendering tasks to some degree.
>
> - Third, our analysis has revealed that instances containing a substantial number of OCR boxes (>5) often pertain to scenarios involving books and newspapers. In such cases, there's a dual challenge: the relatively diminutive size of texts poses difficulty for VAE decoder-based pixel-level reconstruction, and the dense text distribution negatively impacts OCR recognition and Glyph Rendering.
>
> We have visualized some representative cases with more than 5 bounding boxes in **Figure 2 of the attached PDF**. In summary, we wan to relax the restrictions on the Laion-OCR dataset, raising the threshold from 5 to 10 or 20 bounding boxes in the future.
>
> ---
>
> > Q3
>
> A: Great point! We've also noticed that BLIP-2 captions often struggle to consistently depict both image content and OCR text information, resulting in considerable noise. We will attempt the following avenues to address this challenge:
>
> - Employing more potent caption models like LLaVA-from-LLaMA-2 (https://github.com/haotian-liu/LLaVA) and Kosmos-2 (https://github.com/microsoft/unilm/tree/master/kosmos-2). Additionally, we aim to craft specialized prompts to guide the model in generating reliable captions that encompass both image content and OCR text information.
>
> - Enhancing BLIP-2 captions using GPT-3.5 or GPT-4 (text-only) based on supplementary OCR recognition outcomes. Our intention is for the rewritten captions to exhibit a higher quality. We can also explore refining the captions based on the above stronger models such as LLaVA-from-LLaMA-2 and Kosmos-2.
>
> Given the resource-intensive nature of both approaches and the ongoing nature of their implementation, we aspire to incorporate these results into the final revision. We are also keen to receive any further valuable suggestions that could contribute to our progress in this direction.
>
> ---
>
> > Q4
>
> A: Great point! We follow your suggestion to generate images that contain a paragraph rather than a simple phrase or word. We visualize the results in **Figure 2 of the attached pdf**. Our method could preserve the global arrangement following glyph instructions, but fail to generate readable small size text within paragraphs. We would like to further explore this inspiring direction.
>
> ---
>
> > Q5
>
> A: Great suggestion! We concur that editing glyphs (visual text) within an image using text prompts is an immensely valuable avenue to pursue. While our present solution doesn't currently encompass this feature, we are committed to delving into this direction soon. Our plan involves drawing inspiration from previous notable works like Prompt-to-Prompt and Instruct-Pixel2Pixel. We aim to establish a framework in this exciting direction by constructing relevant training data and frameworks.
>
> ---
>
> > Q6
>
> A:  Great point! First, we need to elevate the filtering threshold for the number of boxes and incorporate training data that encompasses a substantial quantity of text. Second, we agree the importance of high-resolution images. Notably, the recently open-sourced SDXL might provide an excellent starting point as it supports resolutions up to 1024x1024. Third, generating high-resolution images surely will bring more computation costs. However, we have observed that the glyph render's efficiency is notably high and only incurs a minor overhead. Furthermore, we are keen to explore various avenues to enhance the efficiency of the diffusion models, a broad and open challenge. This could involve measures like int8 quantization, utilization of advanced samplers like EulerEDMSampler, or even exploring cascade diffusion architectures such as Imagen and its open-source variant, DeepFloyd IF.
>
> ---
>
> > Q7
>
> A: Great point! While text generation might not be essential for all cases, coherence issues can arise from current naive glyph rendering methods, which only support rendering black texts of the same font style on the whiteboard. We're keen to explore advanced font rendering integration, like artistic fonts from Adobe Firefly, within GlyphControl. This approach could work well for poster-like images. Your additional feedback is highly appreciated.
>
> ---
>
> > Q8
>
> A: Your suggestions have been immensely valuable and have significantly enhanced our work. We intend to enrich the training dataset with more OCR boxes, train the models accordingly, and provide both quantitative and qualitative comparison results. We assure you that despite time constraints during the rebuttal period, we'll include these outcomes in the final revision. Your input has played a pivotal role in refining our work.

---

> > ### Comment · Reviewer_xxJ1 · 2023-08-16
> >
> > I appreciate the responses provided by the author during the rebuttal phase. I believe that the author's responses in the feedback should be incorporated into this paper, particularly concerning issues like font consistency and the need for rich text. While these may be considered drawbacks, highlighting these concerns also contributes significantly. Based on the opinions of other reviews and the author's current feedback, I am willing to slightly increase my score.

---

> > > ### Author Response · Authors · 2023-08-17
> > > **Thanks for Reviewer xxJ1's Response**
> > >
> > > We extend our gratitude to reviewer xxJ1 for your valuable comments that enriched our understanding of visual text rendering. We'll incorporate these insights, particularly regarding font consistency and enriched text, in our revision. Your encouragement and increased ratings are greatly appreciated.

---

### Official Review · Reviewer_kTnz · 2023-07-07

**Soundness:** 3 good
**Presentation:** 4 excellent
**Contribution:** 3 good
**Rating:** 6
**Confidence:** 4

**Summary:**

The paper add glyphcontrol to diffusion model by adding controlnet that takes rendered whiteboard images as inputs. Texts in the whiteboard images are extracted by OCR engine during training and then rendered by glyph renderer. During inference, glyph renderer renders the whiteboard images based on the instructions and feed the images using glyph controlnet. They also provide LAION-OCR benchmark by filtering LAION dataset using OCR systems.

**Strengths:**

The incorporation of whiteboard images generated by the glyph renderer into diffusion models for glyph generation, using controlnet, is a reasonable and easy to understand idea. The paper effectively demonstrates the successful synthesis of images with clean glyphs, surpassing the performance of the baselines, and the glyphs can be controlled easily. They also provide LAION-OCR dataset that is useful for OCR-related research.

**Weaknesses:**

The proposed method requires additional training. The method may be seen as an extension of ControlNet and may appear to involve the combination of multiple modules, which could be seen as engineering work.

**Questions:**

Does finetuning with OCR data harm the generation performance of the original model? (visual quality of non-textual parts after fine-tuning)

**Limitations:**

The paper does not explicitly describe their limitations. It would be nice the author provides limitations that can be solved for future works.

---

> ### Author Rebuttal · Authors · 2023-08-10
>
> ## Response to Reviewer kTnz
>
> We thank the reviewer for the careful reviews and constructive suggestions. We answer the questions as follows.
>
> > "The proposed method requires additional training. The method may be seen as an extension of ControlNet and may appear to involve the combination of multiple modules, which could be seen as engineering work."
>
> A: Good point!
>
> First, our method incurs notably lower additional training costs than alternatives like DeepFloyd IF and SDXL. These methods necessitate extensive retraining using thousands of A100 GPUs over months.
>
> Second, GlyphControl combines modules for coherent text-image synthesis (seem like an engineering work to some degree), yet its implementation is both simple and effective. It offers a valuable baseline to address limitations in existing SD models.
>
> Last, we aspire to create concise, innovative frameworks moving forward.  We also would like to learn from your further valuable suggestions.
>
> ---
>
> > "Does finetuning with OCR data harm the generation performance of the original model? (visual quality of non-textual parts after fine-tuning)"
>
> A: No, we believe that finetuning with OCR data using our method would not harm the generation performance of the original model heavily.
>
> When generating **natural images without specifying the text rendering information** (FID-30K-COCO), our model strictly retains the generation abilities of the original stable diffusion model, i.e., **FID evaluated on the COCO dataset would remain the same**. It is because the Controlnet framework does not alter the original SD model weights and the GlyphControl branch would not be used during inference due to empty glyph image input.
>
> In order to test the visual qualities of generating **text images**, We evaluate the FID metrics on LAION-OCR. For LAION-OCR, we select examples which are not included in the dataset for GlyphControl training.
>
> | Method           |   FID-10K-LAION-OCR $\downarrow$ |
> | :---------------- |  :---------: |
> | SDXL-1.0  | $44.77$ |
> | Stable Diffusion v2.1         | $50.01$     |
> | Stable Diffusion v2.0        |  $39.23$  |
> | DeepFloyd (IF-I-M)           | $23.53$ |
> | DeepFloyd (IF-I-L)          |   $30.85$  |
> | DeepFloyd (IF-I-XL)          |  $26.34$ |
> | GlyphControl-SDv2.0 (LAION-OCR-100K)         | $29.13$   |
> | GlyphControl-SDv2.0 (LAION-OCR-1M)       |   $28.02$ |
>
> Our approaches show comparable or slightly worse performances with DeepFloyd IF models in terms of **FID-10K-LAION-OCR**. It implies that the diversity of visual text image generation is preserved by our method and the quality of non-textual parts in the text images would not be harmed heavily.
>
> ---
>
> > "
> The paper does not explicitly describe their limitations. It would be nice the author provides limitations that can be solved for future works."
>
> A: Thanks for your great suggestions. These are some limitations of our method:
> - Lack of controllability on text font, color, and style
> - Poor performance when generating abundant text or characters with small font sizes
> - Using samples with a limited number of OCR boxes for training
>
> We will also include discussions about the limitations and future work in our revised paper.

---

> > ### Comment · Reviewer_kTnz · 2023-08-17
> >
> > As the reviewer wGfp said, I agree that the work is highly motivated by ControlNet, but I think the work is still valuable for OCR community. After reading the rebuttal, I keep my original rate.

---

> > > ### Author Response · Authors · 2023-08-18
> > > **Thanks for Reviewer kTnz's Response**
> > >
> > > We express our gratitude for the positive remarks provided by reviewer kTnz regarding the value of our work within the OCR community. Your encouragement and favorable evaluation are truly cherished.

---

### Official Review · Reviewer_wGfp · 2023-07-12

**Soundness:** 2 fair
**Presentation:** 2 fair
**Contribution:** 2 fair
**Rating:** 4
**Confidence:** 5

**Summary:**

In this paper, a glyph-conditional text-to-image generation model named GlyphControl is proposed for visual text generation. In addition, the authors introduce a visual text generation benchmark named LAION-OCR by filtering the LAION-2B-en. The results show that method of this paper outperforms DeepFloyd IF and Stable Diffusion in terms of OCR accuracy and CLIP score.

**Strengths:**

1.	The visualization results shown in this paper are impressing. The images presented in the paper show that the text of the generated text region is accurate at both the character level and the word level.
2.	The introduction of LAION-OCR dataset is a modest contribution to the realm of NLP and text image generation.


**Weaknesses:**

1.	The contribution is limited. The whole model is the same as ControlNet. The LAION-OCR dataset is just filtered out from an open-source dataset LAION-2B-en, which is not collected by the authors.
2.	The results of the quantitative experiment indicate that the accuracy achieved using this method is merely 40%/26%, which to some extent suggests that the visualized outcomes have been carefully selected. More results need to be displayed.
3.	The comparison is unfair. The compared methods do not prioritize text generation.


**Questions:**

1.	Does the glyph render module only support a single bounding box input or can it accommodate multiple bounding box inputs?
2.	Does the method described in this paper utilize classifier-free guidance during sampling? Although caption dropping is mentioned in the experimental details, there is no specific mention of classifier-free guidance.


**Limitations:**

The manageable elements of users are still constrained. Users are unable to select the color or the font.

---

> ### Author Rebuttal · Authors · 2023-08-10
>
> ## Response to Reviewer wGfp
>
> We thank the reviewer for the careful reviews and constructive suggestions. We answer the questions as follows.
>
> > "The contribution is limited. The whole model is the same as ControlNet. The LAION-OCR dataset is just filtered out from an open-source dataset LAION-2B-en, which is not collected by the authors."
>
> A: Please refer to the general response. In general, the main contribution of this work is  to provide a surprisingly simple yet effective solution to rendering legible (visually coherent) text. We are committed to refining this work further based on any additional valuable suggestions you may provide.
>
> ---
>
> > "The results of the quantitative experiment indicate that the accuracy achieved using this method is merely 40%/26%, which to some extent suggests that the visualized outcomes have been carefully selected. More results need to be displayed."
>
> A: Great suggestion! We have followed your valuable comments to visualize more failure cases and illustrate different types of errors by pointing out the places where the errors occur in **Figure 3 of the attached PDF**. We also would like to include these visualization results and add more analysis in the final revision.
>
> ---
>
> > "The comparison is unfair. The compared methods do not prioritize text generation."
>
> A: We disagree with your statement for the following reasons:
>
> -**Training cost**: The key factors contributing to the success of these models are (i) using much stronger text encoders (DeepFloyd IF uses T5-XXXL with 4.8B parameters, SD-XL uses a combination of open CLIP-G and CLIP-L with 817M parameters) and (ii) re-training the entire text-to-image diffusion models from scratch. These factors require thousands of A100 GPUs for training over several months. Given that our method only requires fine-tuning off-the-shelf models, it is unfair to claim that the comparisons are biased considering the substantial training costs.
>
> -**Position of this work**: As reported in Table 1, both DeepFloyd IF[2] and SD-XL[3] still struggle to render accurate legible text. The Midjourney model performs even worse when handling this challenging task. We want to emphasize that our method is not intended to replace these strong models but to improve their accuracy. We are also making significant efforts to integrate our method into these robust baselines, and we will include these results in the revision, such as GlyphControl + DeepFloyd IF and GlyphControl + SD-XL.
>
> [1] https://huggingface.co/stabilityai/stable-diffusion-2-1#limitations
>
> [2] https://www.deepfloyd.ai/deepfloyd-if / https://github.com/deep-floyd/IF
>
> [3] https://huggingface.co/stabilityai/stable-diffusion-xl-base-1.0
>
> Overall, it's challenging to definitively claim the comparison as unfair given the noted distinctions. We aspire for our work to play a pivotal role in advancing the accuracy of visual text rendering. We would like to hear your further valuable feedback.
>
>
> ---
>
> > "Does the glyph render module only support a single bounding box input or can it accommodate multiple bounding box inputs?"
>
> A: Great point! Indeed, we have effectively demonstrated that the glyph render module is capable of handling multiple bounding box inputs. You can find a detailed illustration in the main paper, precisely in text lines 157-158 and the fourth example in Figure 4. Moreover, all four examples showcased in Figure 1 have been generated using multiple bounding box inputs, further substantiating our capabilities in this aspect.
>
> ---
>
> > "Does the method described in this paper utilize classifier-free guidance during sampling? Although caption dropping is mentioned in the experimental details, there is no specific mention of classifier-free guidance."
>
> A:  Yes, we adopt the classifier-free guidance following other diffusion models like Stable Diffusion and ControlNet. Thanks for pointing out this important detail we have missed and we will explain how we use classifier-free guidance in the revised paper.
>
> ---
>
> > "The manageable elements of users are still constrained. Users are unable to select the color or the font."
>
> A: Thank you for your insightful suggestion! We are committed to delving into the possibilities of controlling colors and fonts in our future endeavors.  We also welcome and appreciate your further valuable suggestions.

---

> > ### Author Response · Authors · 2023-08-18
> > **Looking forward to hearing the feedback from Reviewer wGfp**
> >
> > We extend our sincere gratitude for the invaluable guidance you have offered in the refinement of our work. Our highest priority is to rigorously address the primary concerns you have raised, particularly those related to the perceived limitations of our contributions and the fairness of comparisons.
> >
> > We humbly welcome any additional suggestions you may deem fit to share. Your insights hold immense importance for us, and we view them as integral to the ongoing enhancement of our work.

---

### Official Review · Reviewer_Cb1X · 2023-07-14

**Soundness:** 2 fair
**Presentation:** 3 good
**Contribution:** 2 fair
**Rating:** 4
**Confidence:** 4

**Summary:**

This work proposes GlyphControl for visual text generation by augmenting textual prompt with additional glyph conditional information. A benchmark of LAION-OCR is built for evaluating this model.

**Strengths:**

++The task of visual text generation is interesting.

++Good results are shown in experiments.

++A new visual text generation benchmark is introduced.


**Weaknesses:**

--The main concern is the limited technical contribution. The whole architecture of GlyphControl is a simple extension of ControlNet by using additional control of glyph images. But the idea of glyph images has been validated in GlyphDraw [13] for the same task.

--I am curious by the claim of “outperforms DeepFloyd IF and Stable Diffusion in terms of 55 OCR accuracy and CLIP score while saving the number of parameters by more than 3x”. As shown in Table 1, the parameter number of GlyphControl is significantly larger than Stable Diffusion v2.0.

--Why not train GlyphControl on the common SD 2.1, which has the similar number of parameters and clearly outperforms SD 2.0.

--More competitive baselines should be included for comparison (GlyphDraw [13], SD XL and Midjourney).

--As in GlyphDraw [13], it is necessary to report the FID values in performance comparison.


**Questions:**

Moreo discussion on technical contribution and more comparison with competitive baselines.

---

> ### Author Rebuttal · Authors · 2023-08-10
>
> ## Response to Reviewer Cb1X
>
> We thank the reviewer for the careful reviews and constructive suggestions. We answer the questions as follows.
>
> > "The main concern is the limited technical contribution. The whole architecture of GlyphControl is a simple extension of ControlNet by using additional control of glyph images. But the idea of glyph images has been validated in GlyphDraw [13] for the same task."
>
> A: Please refer to the general response for addressing the concerns regarding the limited technical contribution. To sum up, the main contribution of this work lies in introducing an astonishingly simple yet highly effective approach for rendering legible and visually coherent text.
>
> ---
>
> > "I am curious by the claim of “outperforms DeepFloyd IF and Stable Diffusion in terms of 55 OCR accuracy and CLIP score while saving the number of parameters by more than 3x”. As shown in Table 1, the parameter number of GlyphControl is significantly larger than Stable Diffusion v2.0."
>
> A: Thanks for pointing out the typo, we will fix it in the revised paper. "More than 3x" refers to the comparison with DeepFloyd IF-I-XL (shown below), which achieves the best OCR accuracy among other baselines.
>
> | Method | \#Params | Text Encoder |
> | :---------------- | :--------- | :--------|
> | DeepFloyd (IF-I-XL) | $6.0$ B | T5-XXL ($4.8$ B)|
> | GlyphControl  | $1.3$ B |  CLIP ($354$ M) |
>
> The total number of parameters reported in the second column of the table does not include the text encoder.
>
> ---
>
> > "Why not train GlyphControl on the common SD 2.1, which has the similar number of parameters and clearly outperforms SD 2.0."
>
> A: We chose SD 2.0 to demonstrate the effectiveness of our framework and it could be transferred onto SD 2.1. We are training the GlyphControl framework based on SD 2.1 using the LAION-OCR-100k while keeping the same training settings as the former experiments on SD 2.0. The results of OCR accuracy, CLIP Score, and FID are shown below.
>
> | Method           | Epochs |  $\bf{Acc}$ (%) $\uparrow$ | $\bf{\hat{Acc}}$ (%) $\uparrow$ |$\bf{LD}\downarrow$ | CLIP Score $\uparrow$ |  FID-10K-LAION-OCR $\downarrow$ |
> | :---------------- | :---------: | :---------: | :--------:| :---------: | :---------: | :---------: |
> | GlyphControl-SDv2.0        | 60 | $30/19$  | $37/24$ | $1.77/2.58$  | $33.69/36.20$ | $29.13$ |
> | GlyphControl-SDv2.0        |  30 | $20/14$ | $28/19$  | $2.20/3.00$ | $33.73/36.14$ |  $29.18$ |
> | GlyphControl-SDv2.1        |  30  | $17/14$ | $27/19$  | $2.36/3.18$ | $33.85/35.54$ | $28.02$ |
>
> After training for 30 epochs, the GlyphControl model trained on SDv2.1 shows slightly worse performance compared with the counterpart trained on SDv2.0 previously in terms of OCR accuracy and CLIP score. While the fidelity of images (FID) has been improved, which may be attributed to the more powerful generation abilities of SDv2.1.
>
> And we will report the results of GlyphControl-SDv2.1 in the revised paper after finishing the 60-epoch training.
>
> ---
>
> > "More competitive baselines should be included for comparison (GlyphDraw [13], SD XL and Midjourney)."
>
> A: As we check in the github repo of GlyphDraw [13] https://github.com/OPPO-Mente-Lab/GlyphDraw, the checkpoints of GlyphDraw are not available. Thus, we could not conduct a fair comparsion on our benchmark. Besides, Midjourney is not a free open-sourced tool, thus we fail to make a comparison with Midjourney.
>
> SDXL was not open-sourced until June, 2023, and the latest version SDXL-1.0 has been released at https://github.com/Stability-AI/generative-models right before the rebuttal. The comparison results with SDXL-1.0 are reported here and will be added to the paper. We use both SDXL-base-1.0 and SDXL-refiner-1.0.
>
>
> | Method           |  $\bf{Acc}$ (%) $\uparrow$ | $\bf{\hat{Acc}}$ (%) $\uparrow$ |$\bf{LD}\downarrow$ | CLIP Score $\uparrow$ |  FID-10K-LAION-OCR $\downarrow$ |
> | :---------------- | :---------: | :--------:| :---------: | :---------: | :---------: |
> | SDXL-1.0 | $0.3/0.5$ |  $13/8$ | $6.26/6.30$ | $31.9/33.3$ |  $44.77$   |
> | GlyphControl-SDv2.0 (LAION-OCR-100K)          | $30/19$  | $37/24$ | $1.77/2.58$  | $\bf{33.7}/\bf{36.2}$ |   $29.13$ |
> | GlyphControl-SDv2.0 (LAION-OCR-1M)          | $\bf{40}/\bf{26}$  | $\bf{45}/\bf{30}$ | $\bf{1.59}/\bf{2.47}$  | $33.4/36.0$ | $28.02$ |
>
>
> As seen in the above table, our method still significantly outperforms the latest powerful text-to-image generation model SDXL-1.0 in terms of OCR accuracy, CLIP Score, and FID. Moreover, we also include some visualization results in **Figure 1 of the attached PDF** for comparison with competitive baselines, i.e., IF, SDXL, and Midjourney. And we will add the results of SDXL-1.0 to Table 1 & 2 and include visualization results in the revised paper.
>
> ---
>
> > "As in GlyphDraw [13], it is necessary to report the FID values in performance comparison."
>
> A: In order to test the visual quality of generated text images, We evaluate the FID on our LAION-OCR after selecting examples which are not used to train GlyphControl framework.
>
> | Method           |   FID-10K-LAION-OCR $\downarrow$ |
> | :---------------- |  :---------: |
> | SDXL-1.0  | $44.77$ |
> | Stable Diffusion v2.1         | $50.01$     |
> | Stable Diffusion v2.0        |  $39.23$  |
> | DeepFloyd (IF-I-M)           | $23.53$ |
> | DeepFloyd (IF-I-L)          |   $30.85$  |
> | DeepFloyd (IF-I-XL)          |  $26.34$ |
> | GlyphControl-SDv2.0 (LAION-OCR-100K)         | $29.13$   |
> | GlyphControl-SDv2.0 (LAION-OCR-1M)       |   $28.02$ |
>
> Our approaches show comparable performances with DeepFloyd IF models in terms of FID. This demonstrates that the diversity and quality of visual text image generation is preserved by our method. And both SDXL and original SD models perform much worse than ours.
>
> And we will add FID values into Table 1.

---

> > ### Comment · Reviewer_Cb1X · 2023-08-17
> >
> > I appreciate the additional experimental results in rebuttal. Some concerns have been addressed well. However, for the evaluation on visual quality of generated text images, I am supervised to see DeepFloyd (IF-I-M) outperforms the proposed GlyphControl-SDv2.0, and Stable Diffusion v2.0 even outperforms the recent upgraded version of stable diffusion SDXL-1.0. The results somewhat reveal the weakness of the proposed GlyphControl on visual quality of generated text images.

---

> > > ### Author Response · Authors · 2023-08-17
> > > **Response to Reviewer Cb1X**
> > >
> > > 👉 First, thank you for acknowledging the additional experimental results provided in the rebuttal. We are pleased to note that certain concerns have been effectively addressed.
> > >
> > > 👉 Second, regarding the evaluation of visual quality in generated text images, we acknowledge that we are also surprised to find that DeepFloyd (IF-I-M) outperforms both our GlyphControl-SDv2.0 and the very recent SDXL-1.0.  We assure you that, if necessary, we will **incorporate additional visualization comparison results in the revised version to provide a more comprehensive understanding**. The dominant factor contributing to this outcome may lie in **the utilization of the notably more powerful T5 XXL text encoder (with 4.8B parameters)**. We are enthusiastic about investigating how the combination of GlyphControl and DeepFloyd (IF-I-M) could harmonize their strengths, leading to an enriched system. This direction holds our keen interest for future exploration.
> > >
> > > 👉 Third, these outcomes indeed illuminate potential constraints within the visual quality realm of our GlyphControl. Your perceptive observations are greatly valued, and we intend to meticulously investigate how to enhance visual quality in accordance with your invaluable suggestions.
> > >
> > > 👉 Last, it's worth emphasizing that images exhibiting enhanced quality through **DeepFloyd (IF-I-M) appear to compromise OCR accuracy**. We speculate that **a trade-off between elevated OCR precision and visual quality emerges when tackling such a demanding visual text generation task**. Furthermore, it's important to note that **DeepFloyd (IF-I-M) incorporates a significantly larger number of parameters compared to our approach (6.9B vs. our 1.65B)**. To provide a comprehensive reference, we present the complete set of comparison results below:
> > >
> > > | Method           |  \# Overall Params (# Text Encoder) |  $\bf{Acc}$ (%) $\uparrow$ | $\bf{\hat{Acc}}$ (%) $\uparrow$ |$\bf{LD}\downarrow$ | CLIP Score $\uparrow$ |  FID-10K-LAION-OCR $\downarrow$ |
> > > | :---------------- | :---------------- | :---------: | :--------:| :---------: | :---------: | :---------: |
> > > | DeepFloyd (IF-I-M)           | 6.9B (4.8B) | $0.3/0.1$  | $18/11$ | $2.44/3.86$  | $32.8/34.3$ | $\bf{23.53}$ |
> > > | GlyphControl-SDv2.0 (LAION-OCR-100K)   | 1.65B (354M)      | $30/19$  | $37/24$ | $1.77/2.58$  | $\bf{33.7}/\bf{36.2}$ |   $29.13$ |
> > > | GlyphControl-SDv2.0 (LAION-OCR-1M)      | 1.65B (354M)  | $\bf{40}/\bf{26}$  | $\bf{45}/\bf{30}$ | $\bf{1.59}/\bf{2.47}$  | $33.4/36.0$ | $28.02$ |
> > >
> > > Your continued valuable feedback would be greatly appreciated.  🤗🤗🤗

---

> > > > ### Author Response · Authors · 2023-08-18
> > > > **Looking forward to hearing the feedback from Reviewer Cb1X**
> > > >
> > > > Your dedicated guidance has been incredibly valuable in refining our work, and we sincerely appreciate it. We are eagerly seeking to ensure that our responses adequately address your primary concerns, particularly with regard to the visual quality of the generated text images.
> > > >
> > > > We would greatly appreciate any additional suggestions you may have to offer. Your insights are highly valued and considered integral to our work's improvement.

---

### Author Rebuttal · Authors · 2023-08-10

## To AC and All Reviewers


We would like to express our gratitude to all the reviewers for their careful reviews and constructive suggestions. We appreciate the positive comments, such as "the task of visual text generation is interesting" (Reviewer Cb1X), "the visualization results shown in this paper are impressing" (Reviewer wGfp), "reasonable and easy to understand" (Reviewer kTnz), "demonstrate the paper's practical contributions to the field of visual text generation" (Reviewer xxJ1), and "I believe this work is a nice addition to the current research landscape of text-to-image generation" (Reviewer Yxcw).

We would like to address the major concerns regarding the contribution of this work by focusing on the following aspects:

> **The motivation and potential impact of our work**

First, we would like to emphasize that addressing the well-known limitations[1] of rendering legible (visually coherent) text is a significant challenge for the fundamental Stable-Diffusion model series.

Second, we acknowledge that some groundbreaking works, such as DeepFloyd IF[2] and SD-XL[3], which emphasize their strong capabilities of rendering legible text, were made public shortly before or after our submission. However, we disagree with Reviewer wGfp's statement that "the comparison is unfair, the compared methods do not prioritize text generation," for the following reasons:

-**Training cost**: The key factors contributing to the success of these models are (i) using much stronger text encoders (DeepFloyd IF uses T5-XXXL with 4.8B parameters, SD-XL uses a combination of open CLIP-G and CLIP-L with 817M parameters) and (ii) re-training the entire text-to-image diffusion models from scratch. These factors require thousands of A100 GPUs for training over several months. Given that our method only requires fine-tuning off-the-shelf models, it is unfair to claim that the comparisons are biased considering the substantial training costs.

-**Position of this work**: As reported in Table 1, both DeepFloyd IF[2] and SD-XL[3] still struggle to render accurate legible text. The Midjourney model performs even worse when handling this challenging task. We want to emphasize that our method is not intended to replace these strong models but to improve their accuracy. We are also making significant efforts to integrate our method into these robust baselines, and we will include these results in the revision, such as GlyphControl + DeepFloyd IF and GlyphControl + SD-XL.

[1] https://huggingface.co/stabilityai/stable-diffusion-2-1#limitations

[2] https://www.deepfloyd.ai/deepfloyd-if / https://github.com/deep-floyd/IF

[3] https://huggingface.co/stabilityai/stable-diffusion-xl-base-1.0

> **Comparison with GlyphDraw and ControlNet**

First, we would like to clarify that **GlyphDraw is a concurrent work made available on arXiv on 31st March 2023**, which explores the benefits of glyph images in a distinct manner.

Second, we want to emphasize that the ControlNet architecture design is not our contribution. Our contributions lie in (i) demonstrating that using glyph images is a surprisingly simple yet highly effective approach for generating legible text, and (ii) introducing the LAION-Glyph benchmark to facilitate the development of this challenging generation task.

Thirdly, we outline the key differences between our GlyphControl and GlyphDraw:

- Our GlyphControl offers more flexible controllability than GlyphDraw by supporting customized glyph instructions, allowing for control over text line information and text box information (as evidenced by the visual results in Figure 4).
- GlyphDraw requires fine-tuning the cross-attention weights within the U-Net, which may harm the generation capability of the original diffusion model. In contrast, our GlyphControl follows the ControlNet scheme, freezing the well-trained U-Net weights to preserve the original model's capability optimally.
- GlyphDraw necessitates (i) an additional mask prediction module to predict segmentation masks, and (ii) a CLIP image encoder to extract visual representations of the corresponding glyph image. Our GlyphControl, however, only requires an additional Glyph ControlNet to constrain the latent representations within the U-Net.

Lastly, we would like to express our gratitude for the positive and accurate comments from Reviewer Yxcw: **"This work provides a simple yet effective approach to this challenge by elegantly extending ControlNet with rendered text as control. I am convinced that having rendered text as input makes a lot of sense in this context and would greatly help generate accurate text."** We sincerely hope the reviewers will reconsider the original ratings.

PS: **Please refer to the PDF for more visualization results.**

---

### Author Response · Authors · 2023-08-12
**Looking forward to hearing the response from all Reviewers**

We appreciate Reviewer Yxcw's swift response and have gained valuable insights from the constructive reviews provided by other reviewers.

Your meticulous evaluations and invaluable suggestions are highly regarded, contributing significantly to the improvement of our submission.

We eagerly await additional suggestions and are committed to offering necessary responses to address any concerns you may have. 🤗🤗🤗

---

### Comment · Area_Chair_zhEZ · 2023-08-15
**Please read the authors' responses and start discussing.**

Hi reviewers,

The authors have responded to your reviews. Please carefully read their responses, consider to what extent they address your concerns, and update your review accordingly. If you have follow-up questions to anything the authors have written in their responses, I encourage you to reply to them and continue the discussion.

Your AC

---

### Author Response · Authors · 2023-08-21
**Thanks to All Reviewers and AC**

## To All Reviewers and AC

👉 We extend our sincere appreciation for your thoughtful evaluation of our work and the invaluable insights you've shared. Your feedback has been instrumental in refining our research. As we approach the final phase of the discussion period, we wish to express our commitment to addressing your concerns comprehensively in our rebuttal.

👉 We are eager to hear the feedback from Reviewer Cb1X and Reviewer wGfp. Should you find it necessary, we are fully prepared to embark on further clarification or additional experimentation, with the aim of enhancing the robustness of our study. Your guidance is pivotal to our endeavor, and we are dedicated to incorporating your suggestions into the paper's revision process.

👉We are deeply grateful for the time and effort you have generously invested in reviewing our work. Your dedication is truly commendable, and we encourage you to reach out with any remaining questions or suggestions.

Thank you once again for your invaluable contributions!

---

### Decision · Program_Chairs · 2023-09-21

**Decision:**

Accept (poster)

**Comment:**

How to generate high-quality and accurate glyphs in the text2image task is challenging. This paper proposed an effective and simple method to handle this problem. The reviews of this paper are slightly mixed. Two reviewers gave “Borderline reject” while three reviewers were very positive on the paper (2 “Weak Accept” and 1 “Accept”). After carefully reading the authors’ rebuttal and all reviewers’ comments. The AC agrees with the majority of reviewers that this paper has made valuable contributions to this challenging task and showed impressive synthesis results. This decision was discussed with and approved by the SAC.